# GroundingDINO for Open-Set Lesion Detection in Medical Imaging

Samuel J. Roughley[1], Johanna P. Müller[2], Shangqi Gao[3,4], Zeyu Gao[3,4], Marta Ligero[5], Rudolfs Blums[6], Mireia Crispin-Ortuzar[3,4], Julia Schnabel[6,7,8,9], Bernhard Kainz[2,10], Cosmin I. Bercea[6,7], and Ines Prata Machado[3,4]

[1] Department of Physics, University of Cambridge, UK
[2] Friedrich-Alexander University Erlangen-Nürnberg, Erlangen, Germany
[3] Department of Oncology, University of Cambridge, UK
[4] Early Cancer Institute, University of Cambridge, UK
[5] Else Kroener Fresenius Center for Digital Health, Technical University Dresden, Dresden, Germany
[6] Technical University of Munich, Munich, Germany
[7] Helmholtz AI and Helmholtz Center Munich, Germany
[8] Munich Center for Machine Learning, Germany
[9] King's College London, UK
[10] Imperial College London, UK

**Abstract.** Open-world anomaly detection is a task in which machine learning is well-positioned to advance cancer diagnosis, potentially leading to significantly improved survival rates. For a model to be used in clinical settings, it must demonstrate high performance, robustness, and generalisability. A common approach to achieving high generalisability is to incorporate information from broader representations within the model. In this work, we investigate the application of GroundingDINO to medical anomaly detection and localisation, evaluating both its overall performance and the influence of text prompts. We find that GroundingDINO outperforms the YOLOv11n model even with minimal use of contextual information. When exploring methods to introduce more contextual information, we observe that specifying the organ within the prompt improves closed-set performance on rarer lesion classes. However, adding visual descriptions of lesions during training leads to a significant performance drop on those subsets, indicating that the model memorises prompt-image pairs rather than learning meaningful semantic relationships. Our work highlights a critical limitation of GroundingDINO in medical imaging and proposes targeted modifications to the model architecture or training strategies as promising directions for utilising richer semantic prompts to improve anomaly detection.

**Keywords:** Anomaly Detection · GroundingDINO · Prompt Engineering · Medical Imaging · Lesion Detection · Cancer Research

## 1   Introduction

Early detection is critical to improving survival outcomes for cancer, which accounts for nearly 1 in 6 deaths globally [16][14]. To aid in diagnosis, medical imaging technologies such as Computed Tomography (CT) and Magnetic Resonance Imaging (MRI) provide detailed 3D anatomical views. However, automated identification of open-world anomalies in these scans has not kept pace with advancements in imaging technologies, as interpreting the resulting images remains highly challenging [17]. For example, studies have found that approximately one-third of diagnoses are often missed across various diagnostic pathways [8][3]. Therefore, research into computer-aided cancer detection is invaluable not only for improving cancer survival rates but also for alleviating the growing burden on healthcare systems. As such, significant effort has been dedicated to developing machine learning models for medical anomaly detection (AD). The appearance of cancer varies widely across types, subtypes, and individual patients, making robust open-set performance challenging [7][20]. However, for a model to be clinically viable, it must be capable of detecting both common and rare, or previously unseen, pathologies. A common strategy for improving generalisability is to incorporate contextual information into the model. For example, the GroundingDINO model achieves state-of-the-art open-set performance in the natural imaging domain by introducing language prompts into a closed-set detector [11]. Despite such successes, however, these methods remain relatively underexplored in the medical domain. **Contributions.** We present the first investigation of GroundingDINO for medical anomaly detection, focusing on lesion detection in CT scans of the chest–abdomen–pelvis region, and compare its performance with the state-of-the-art YOLOv11n model. Through a series of experiments using varied text prompts, we examine the impact of prompt design on both closed-set and open-set performance, exploring how semantic information can enhance medical AD. Our ultimate goal is to lay the groundwork for future integration of text and image modalities to achieve state-of-the-art performance with real clinical applicability.

## 2   Methodology

**Background.** GroundingDINO is a transformer-based vision–language model originally trained for object detection on natural images. Its primary goal is to generalise to unseen object classes by integrating semantic information via language into the closed-set detector DINO [21], thereby enabling open-set capabilities. The model's architecture includes three cross-modality fusion points, which the authors argue provide stronger language guidance during detection compared to models with fewer fusion locations [11]. Open-set detection is particularly relevant in medical imaging tasks such as cancer screening, where rare and previously unseen lesions may be encountered. Recent work has highlighted the importance of integrating semantic priors to improve detection generalisation in these settings [2]. Recent advances in Large Language Models (LLMs),

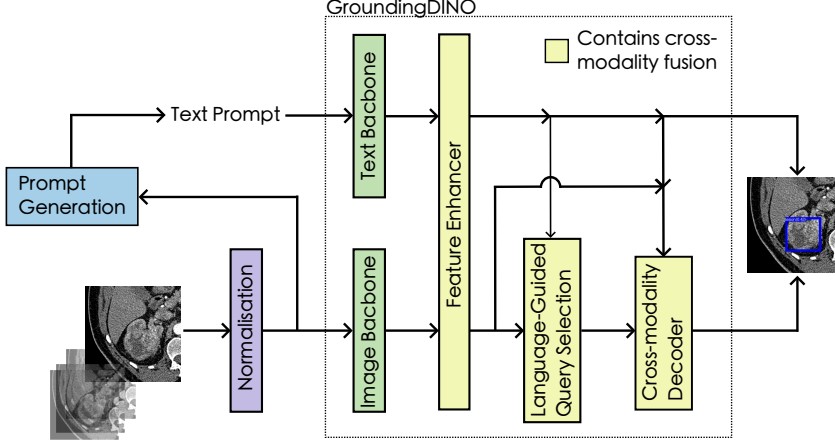

**Fig. 1.** Detection pipeline used during experiments, highlighting the inclusion of the GroundingDINO architecture. A single slice is normalised and a text prompt generated, before being passed to GroundingDINO to perform the detection. The locations of cross-modality fusion are highlighted: cross-attention blocks within the feature enhancer and decoder, and language guidance of query selection. Multiple methods for prompt generation were explored, so it is shown generally.

such as Gemini [18], BiomedGPT [13], and ChatGPT-4 [1], have demonstrated strong capabilities in generating clinically rich, context-aware descriptions. These models provide a powerful means of constructing descriptive prompts to guide open-set detection models in medical applications [12]. Despite its comparatively modest size and training data, GroundingDINO achieves state-of-the-art performance on open-set detection benchmarks, outperforming larger models such as GLIP [9] in the COCO zero-shot setting [10]. Its utility in medical contexts has already been demonstrated in the BiomedParse study [22], where it was used to propose bounding boxes without additional training.

**Model Architecture.** The pipeline used in our experiments is illustrated in Figure 1. Since GroundingDINO is limited to 2D detection, a single slice must first be selected from the scan. The slice is then normalised to improve consistency across samples. Before being passed to GroundingDINO, a text prompt must also be generated. As the method of prompt generation varies across our experiments, a general representation is shown in Figure 1. When relevant, the images are used post-normalisation to generate the prompts. The prompt and normalised image are then passed to the GroundingDINO architecture, where the text and image backbones extract features from the inputs. The feature enhancer then updates the features, making use of text-to-image and image-to-text cross-attention. The updated text features then guide the selection of queries to be used in the decoder, where text and image cross-attention are used to generate the model outputs. For additional details, we refer the reader to the original work by Liu et al. [11]

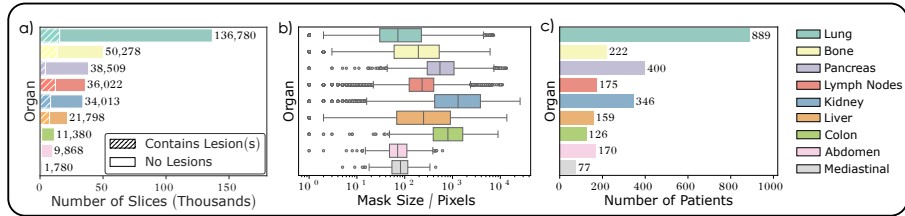

**Fig. 2.** Breakdown of the ULS23 Dataset. a) Number of slices from scans containing (no) lesions in each organ. b) Distribution of mask sizes by organ, with samples outside 1.5 times the IQR from the nearest quartile shown as outliers. c) Number of patients with scans of lesions in each organ.

## 3   Evaluation

**Datasets.** For training and evaluation, we used the Universal Lesion Segmentation Challenge 2023 (ULS23) dataset [5], comprising chest–abdomen–pelvic CT scans with segmentation mask annotations. The dataset contains $6,382$ lesions from $2,627$ patients across various organs (Figure 2). Each scan is cropped to a volume of interest (VOI) of $256 \times 256 \times 128$ voxels, centred on a single annotated lesion. Although lesion centring introduces bias, this controlled setup establishes baseline performance. Extending to whole-volume detection is required for clinical use and can be addressed in future translation work.

**Pre-processing.** Annotations of multiple lesions from the same scan were combined into a single annotation without merging adjacent lesions, enabling detection use. Segmentation masks were converted to bounding boxes for GroundingDINO. Scans were normalised first. Due to wide variation in Hounsfield units (HU) across lesions, fixed windowing was unsuitable. Following the ULS23 baseline, Z-score normalisation was applied per slice. Visibility—measured as the absolute difference between median lesion and surrounding intensities divided by local standard deviation—improved for all lesion types except those in bone, indicating potential bias. The dataset was split into 80 % training (274,617 slices), 10 % validation (33,995 slices), and 10 % testing (36,230 slices), with organ-specific and patient-level separation to prevent data leakage.

**Experiments.** Three experiment types were conducted with models trained on prompts of varying detail. The first used a simple prompt, *"lesion"*, for all scans, providing minimal language guidance and serving as a baseline. Equivalent YOLOv11n models [6], which lack language input, were trained for comparison, mainly relevant to this first experiment. The second experiment specified the organ in the prompt (e.g., *"[organ] lesion"*). The third fine-tuned these models using visual descriptions generated by Gemini ('gemini-2.5-pro-preview-03-25' model) [18], focusing on lymph node lesions due to their moderate sample size and lower initial performance. All three experiments were run both with all lesion types and with mediastinal lesions ($4,879$ training slices) excluded, as they had the fewest samples, minimising training set reduction. Testing on excluded me-

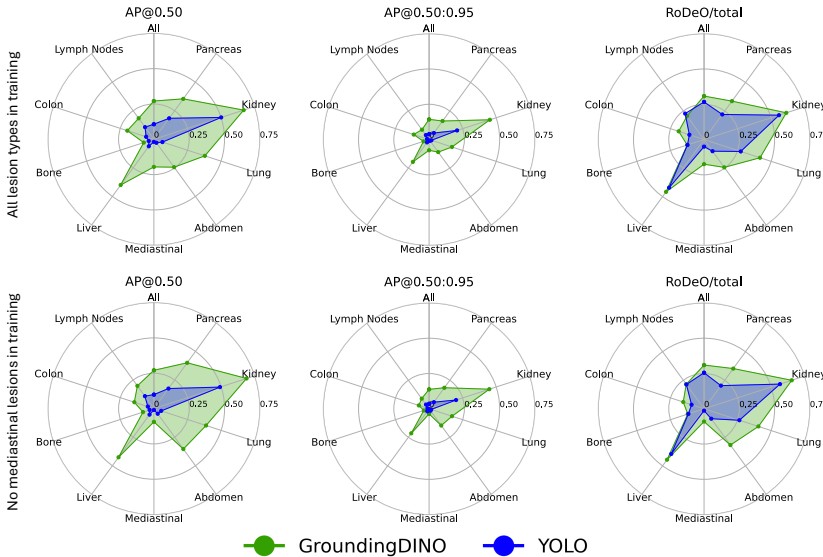

**Fig. 3.** Radar charts comparing the performance of GroundingDINO and YOLO stratified by organ, with the prompt of *"lesion"* given to GroundingDINO. Average Precision and RoDeO/total metrics are shown for GroundingDINO (green) and YOLO (blue) models that saw all lesion types (top) and all except mediastinal lesions (bottom) from the ULS23 dataset during training.

diastinal lesions evaluates open-set performance. Since data consists of cropped CT scans, each shows only a small anatomical region.

**Training Strategy.** To train GroundingDINO, the Open-GroundingDINO training code was used with default model hyperparameters and data augmentations [23]. The released GroundingDINO model with the Swin-T image backbone was used as the initialization, and bert-base-uncased [4] from Hugging Face [19] served as the text backbone. For YOLO training, the default implementation from the Ultralytics Python package was used. All models were trained for 25 epochs on NVIDIA A40 and L40S GPUs. To evaluate model performance, we used the Average Precision (AP) and RoDeO [15] metrics. For RoDeO, a bounding box threshold of 0.2 was selected based on sweeps over the validation set.

## 4   Results

### 4.1   Minimal Language Guidance

The results for the GroundingDINO models using the prompt *"lesion"* for all scans, along with the corresponding YOLO models, are shown in Figure 3. GroundingDINO performs as well as or better than YOLO across all organs. GroundingDINO's superior performance using only simple prompts indicates that semantic alignment, not present in YOLO, offers tangible benefits independent of prompt complexity, highlighting the model's potential suitability for

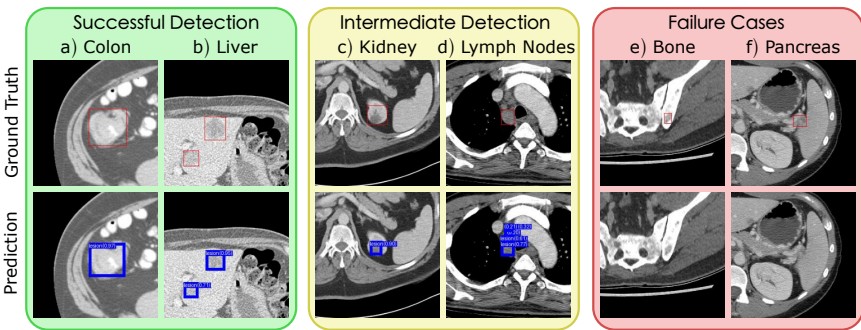

**Fig. 4.** Inference examples from the GroundingDINO model trained on lesions across all organs in the ULS23 dataset, using *"lesion"* as the text prompt. Ground truth annotations (top, red boxes) and model predictions (bottom, blue boxes) are shown for six organ sites.

medical anomaly detection and supporting its use in research such as ours. Inference examples from the GroundingDINO model trained on all lesion types are shown in Figure 4, illustrating both successful detections and failure cases. Figure 4c highlights ambiguities in lesion definition, bounding an internal substructure within the ground truth. Figure 4d contains false positives, typically observed near anatomical features resembling lesion morphology (e.g., vessels or bones). The issue of false positives is noted in the original GroundingDINO paper [11]. The persistence of these issues with minimal prompts points to the need for more precise annotations and improved semantic grounding. As expected, after removing mediastinal lesions from training, performance on mediastinal lesions drops significantly. However, while YOLO's performance falls to near zero (e.g., RoDeO/total = 0.013), GroundingDINO maintains better performance. This better preservation of accuracy, even before introducing additional language guidance, suggests stronger inherent generalisability, making results especially relevant in discussions of clinical deployment. Nevertheless, the sizeable performance drop underscores that open-set detection remains a significant challenge. Consequently, with multimodal models like GroundingDINO, it is natural to consider whether language guidance can mitigate this decline.

### 4.2   Enhanced Language Guidance

The results for the different GroundingDINO models using the three different prompt types are shown in Figure 5.

**Organ-specific prompts.** Organ-specific prompts show no definitive overall effect on performance. A slight improvement is seen when all organs are included in training, but its small magnitude and disappearance when mediastinal lesions are excluded make its significance unclear. Notably, performance improves for colon (54 % RoDeO/total), mediastinal (87 %), and abdominal (30 %) lesions when all lesion types are included. These gains likely result from limited training

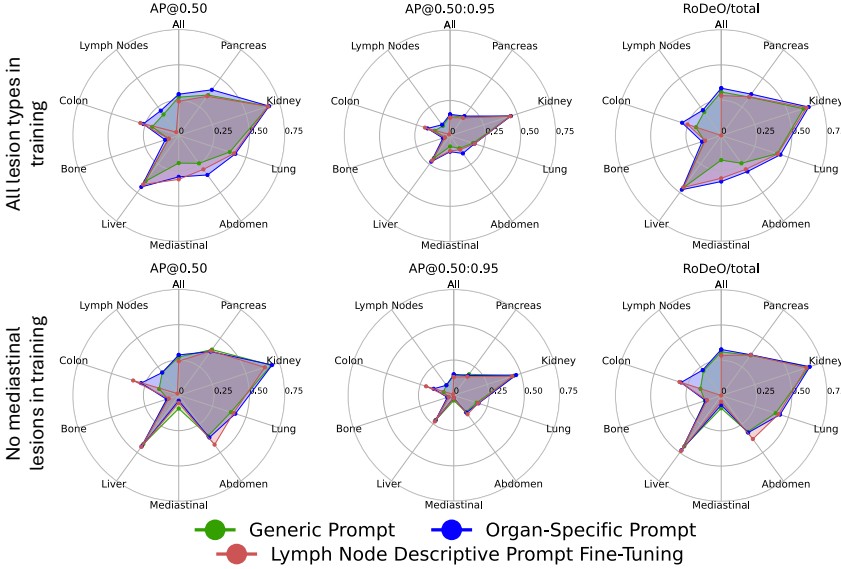

**Fig. 5.** Radar charts comparing the performance of GroundingDINO models stratified by organ, with the models differing by the choice of text prompt used. Average Precision and RoDeO/total metrics are shown for GroundingDINO models that saw all lesion types (top) and all except mediastinal lesions (bottom) from the ULS23 dataset during training. Prompts of *"lesion"* (green), *"[organ] lesion"* (blue) and the addition of visual descriptions (red) were all tested.

data for these lesion types (Figure 2a), making them more susceptible to being overshadowed. Organ-specific prompts reduce interclass competition, helping the model learn relevant visual features. A similar improvement is observed for colon lesions when mediastinal lesions are excluded. However, no gains are seen for mediastinal or abdominal lesions. For mediastinal lesions, this is expected, as the model had no exposure to them. For abdominal lesions, the absence of improvement suggests their performance was suppressed specifically by the presence of mediastinal lesions, despite the latter being the smallest class.

**Descriptive prompts.** After fine-tuning models using visual descriptions for lymph node lesions during training, performance on lymph node lesions dropped to zero. In the test set, none of the model's predictions for lymph node lesions exceeded a confidence score of 0.05, explaining why RoDeO/total = 0. To better understand this behaviour, the model outputs were analysed in more detail. During inference, GroundingDINO generates 900 (box, caption) pairs. For each pair, an activation score is computed for every token in the text input, and tokens with scores above a threshold form the caption. Examples of the mean activation scores across the 900 predictions for a training and testing sample are shown in Figure 6. Activation is highly uniform across tokens. Excluding start and end markers, the maximum activation difference per prediction is just 0.0001 in training and 0.00004 in testing. GroundingDINO is meant to align text and image

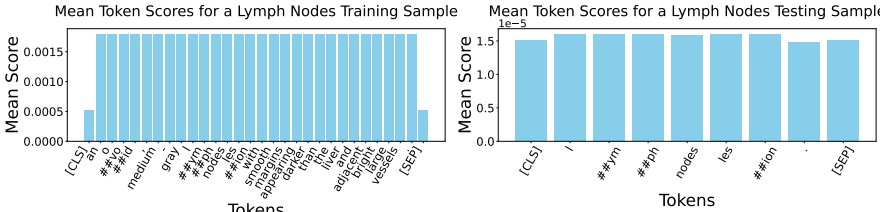

**Fig. 6.** Token-level activation maps from GroundingDINO for a lymph node lesion sample from the training and testing sets, showing uniformly distributed attention across tokens.

features so semantics guide detection, but the uniformity suggests overfitting: the model aligns the entire prompt with image features rather than understanding it. As a result, it fails to link the test prompt "lymph node lesion" to relevant training visuals, leading to inaccurate predictions, especially when training and test prompts differ, which earlier experiments did not reveal. The drop in lymph node performance to zero, despite previous success with descriptive prompts, suggests the initial learning rate was too high. A lower rate might have preserved some understanding but would not fix the uniform activation, which stems from how GroundingDINO learns. Addressing this may require changes to the loss function, text encoder, or prompt engineering.

## 5   Conclusions

GroundingDINO was found to outperform the YOLOv11n model when prompted with the term *"lesion"*, highlighting its suitability for research into medical anomaly detection. Incorporating organ-specific information into text prompts significantly improves closed-set performance on rare lesion classes, emphasising the importance of semantic conditioning. Although overall and open-set performance remain unchanged, these findings suggest clear opportunities for improvement. Using detailed visual lesion descriptions during training revealed overfitting issues that hinder semantic generalization, underscoring the need to refine training methods to better leverage language-based cues.

**Acknowledgments.** C.I.B. is funded via the EVUK programme ("Next-generation AI for Integrated Diagnostics") of the Free State of Bavaria and partially supported by the Helmholtz Association [Munich School for Data Science]. This work is also supported by the Berdelle-Stiftung [TimeFlow]. The authors acknowledge scientific support and HPC resources from NHR@FAU [b143dc, b180dc], funded by federal and Bavarian authorities, with partial hardware funding from the DFG [440719683]. Additional support was received from the ERC [101083647], the DFG [KA 5801/2-1, INST 90/1351-1], and the state of Bavaria. Further funding was provided by Cancer Research UK [A22905], the CRUK Cambridge Centre [CTRQQR-2021-100012, A25177], The Mark Foundation for Cancer Research [RG95043], GE HealthCare, the CRUK National Cancer Imaging Translational Accelerator [A27066], and the NIHR Cambridge Biomedical Research Centre [NIHR203312, BRC-1215-20014].

**Disclosure of Interests.** The authors have no competing interests to declare that are relevant to the content of this article.

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
