# OpenReview forum: "GroundingDINO for Open-Set Lesion Detection in Medical Imaging"
_MICCAI.org/2025/Workshop/MSB_EMERGE — MSB EMERGE 2025 Oral_

### Official Review · Reviewer_iM1m · 2025-07-02

**Recommendation:** 3
**Confidence:** 3

**Clarity:**

The paper is clear and well-written, with minor areas for improvement in clarity

**Feedback:**

Those things are very minor and some are up for discussion:
- You can combine multiple citations in one bracket as [12,13] by writing \cite{abc1,cdf2}
- In Figure 1, you might add three options for your different prompting mechanisms. Also, the arrow merging into the arrow going into the top of the Cross-modality Decoder looks weird in my opinion.
- Somewhere, maybe even in Fig. 1, you can add a few exemplary prompts generated by your prompt generation mechanism.
- I feel like the first sentence in the abstract sounds odd.
- In Fig. 2b the x-axis could be named Mask Size (in Pixels)
- I would structure the prompting mechanisms in the Experiments section using paragraphs at least

**Justification:**

The open-set detection does not work particularly well and judging the method would require more comparisons with baseline methods. Hence, I would recommend to reject the paper.

**Reproducibility:**

Some amount of details available for reproducing the main results, and open access details are unclear

**Strengths:**

- I like Figures 1, 2, and 4 very much.
- The entire manuscript is well written, making the method easy to understand.
- The paper addresses an interesting and important topic of open-set recognition.
- GroundingDINO outperforms the baseline on all 9 anatomies.

**Summary:**

The authors propose a method for open-set detection of lesions based on GroundingDINO

**Weaknesses:**

- The extraction of the descriptive prompt is unclear to me. The CT slice is given to the Gemini model, which generates a textual description of the image. What do the descriptions look like? A few examples would be interesting. Also, I wonder why that should increase the performance of the model. In my opinion, the pretrained LLM will not be able to provide any information that could not be learned by GroundingDINO itself.
- When the prompt "lesion" is given to the model, what is the methodological difference from YOLO other than the changed architecture?
- A comparison with other SOTA detection methods is missing, which would more clearly show if the advantage of GroundingDINO is due to its architecture or the prompting mechanism.
- The volume of interest is centred around the lesion, which might introduce a bias, rendering clinical application more difficult.

---

### Official Review · Reviewer_RnK4 · 2025-07-03

**Recommendation:** 4
**Confidence:** 4

**Clarity:**

The paper is generally clear but has some clarity issues that could be addressed with moderate revision

**Feedback:**

The paper makes a good case for GroundingDINO’s potential in medical anomaly detection, with strong results showing it outperforms YOLOv11 even with minimal prompts. The analysis of overfitting with descriptive prompts is a standout, pinpointing uniform token activation as a key issue.

**Justification:**

The paper is clear with a good main idea, fitting the workshop well. It’s not super new, but still adds useful thoughts.

**Reproducibility:**

Some amount of details available for reproducing the main results, and open access details are unclear

**Strengths:**

- The paper is the first to apply GroundingDINO to medical anomaly detection, specifically lesion detection in CT scans.
- The detection pipeline is well-described, with a logical flow from slice selection to normalization, prompt generation, and detection.
- The radar charts comparing GroundingDINO and YOLOv11 across organs are superb!
- The paper is well-written, easy to follow and understand.

**Summary:**

This paper explores the application of GroundingDINO for open-set lesion detection in chest-abdomen-pelvic CT scans, using the ULS23 dataset. The authors compare GroundingDINO’s performance against the YOLOv11 model, focusing on the impact of text prompt engineering (minimal, organ-specific, and descriptive prompts) on closed-set and open-set detection. Key findings include GroundingDINO’s superior performance over YOLOv11 with minimal prompts, improved closed-set performance for rare lesions with organ-specific prompts, and a significant performance drop when using descriptive prompts due to overfitting.

**Weaknesses:**

- The paper mentions merging annotations of different lesions from the same scan to create a single, comprehensive annotation for detection tasks, but it doesn’t explain how this merging was done. Were overlapping lesions combined into a single bounding box? Were multiple boxes retained per scan?
- Figure 4 shows inference examples, including failure cases like false positives and ambiguous bounding of internal substructures. However, the paper doesn’t analyze why these failures occur beyond citing the original GroundingDINO paper. Are they due to dataset ambiguities, model limitations, or prompt design? This leaves the reader hanging.
- The paper lacks a proper comparison with recent SOTA methods.

---

### Official Review · Reviewer_LbAZ · 2025-07-10

**Recommendation:** 4
**Confidence:** 3

**Clarity:**

The paper is generally clear but has some clarity issues that could be addressed with moderate revision

**Feedback:**

Overall, this is a very relevant work, but the intuition behind performing some of the steps needs to be detailed out.

**Justification:**

Please see my comments above

**Reproducibility:**

Sufficient amount of details available for reproducing the main results, but open access is not provided to source code and/or data

**Strengths:**

1. The paper focuses on an important topic of lesion detection in the context of medical imaging.
2. The research investigates the impact of different text prompts on model performance, specifically identifying that incorporating organ-specific information into prompts significantly improves closed-set performance for rarer lesion classes.
3. The paper shows that GroundingDINO outperforms the YOLOv11n model when prompted with "lesion," highlighting its potential for medical anomaly detection.

**Summary:**

This paper investigates the application of Grounding DINO for lesion detection and localization, specifically focusing on CT scans of the chest, abdomen and pelvis region.

**Weaknesses:**

1. The improvement in the performance of the model is not clear. The slight overall improvement observed when all organs were included during training was not established as significant due to its small magnitude and absence when mediastinal lesions were excluded. My major concern is what is the significance of these improvements?
2. The authors mention that they have fewer training samples for some lesion types, like mediastinal lesions. To address this, the authors excluded them in some experiments to test open-set capabilities but the small sample size for certain categories might inherently limit the model's ability to learn robust features for those specific lesions. So is this exclusion justified?
3. One observation from the results that I notice is that using detailed visual descriptions of lesions during training leads to a significant performance drop. Why this might be the case?
4. It would be good to have some comparison with the state of the art methods to see how well the proposed model performs.